# Progress in the Field of Micro-Electrocorticography

**DOI:** 10.3390/mi10010062

**Published:** 2019-01-17

**Authors:** Mehdi Shokoueinejad, Dong-Wook Park, Yei Hwan Jung, Sarah K. Brodnick, Joseph Novello, Aaron Dingle, Kyle I. Swanson, Dong-Hyun Baek, Aaron J. Suminski, Wendell B. Lake, Zhenqiang Ma, Justin Williams

**Affiliations:** 1Department of Biomedical Engineering, University of Wisconsin-Madison, Madison, WI 53706, USA; mehdi.snm@bme.wisc.edu (M.S.); skkorinek@wisc.edu (S.K.B.); novello@wisc.edu (J.N.); wpvmqor@gmail.com (D.-H.B.); suminski@neurosurgery.wisc.edu (A.J.S.); 2Department of Neurosurgery, School of Medicine and Public Health, University of Wisconsin-Madison, Madison, WI 53792, USA; kiswanson@gmail.com (K.I.S.); lake@neurosurgery.wisc.edu (W.B.L.); 3Department of Electrical and Computer Engineering, University of Wisconsin-Madison, Madison, WI 53706, USA; dwpark31@uos.ac.kr (D.-W.P.); yhjung89@gmail.com (Y.H.J.); 4School of Electrical and Computer Engineering, University of Seoul, Seoul 02504, South Korea; 5Division of Plastic and Reconstructive Surgery, Department of Surgery, University of Wisconsin-Madison, Madison, WI 53792, USA; dingle@surgery.wisc.edu

**Keywords:** electrocorticography, ECoG, micro-electrocorticography, µECoG, neural electrode array, neural interfaces, electrophysiology, brain–computer interface, in vivo imaging, tissue response, graphene

## Abstract

Since the 1940s electrocorticography (ECoG) devices and, more recently, in the last decade, micro-electrocorticography (µECoG) cortical electrode arrays were used for a wide set of experimental and clinical applications, such as epilepsy localization and brain–computer interface (BCI) technologies. Miniaturized implantable µECoG devices have the advantage of providing greater-density neural signal acquisition and stimulation capabilities in a minimally invasive fashion. An increased spatial resolution of the µECoG array will be useful for greater specificity diagnosis and treatment of neuronal diseases and the advancement of basic neuroscience and BCI research. In this review, recent achievements of ECoG and µECoG are discussed. The electrode configurations and varying material choices used to design µECoG arrays are discussed, including advantages and disadvantages of µECoG technology compared to electroencephalography (EEG), ECoG, and intracortical electrode arrays. Electrode materials that are the primary focus include platinum, iridium oxide, poly(3,4-ethylenedioxythiophene) (PEDOT), indium tin oxide (ITO), and graphene. We discuss the biological immune response to µECoG devices compared to other electrode array types, the role of µECoG in clinical pathology, and brain–computer interface technology. The information presented in this review will be helpful to understand the current status, organize available knowledge, and guide future clinical and research applications of µECoG technologies.

## 1. Introduction

Multichannel neural interfaces provide a direct communication pathway between the central nervous system and the ex vivo environment. These front-end devices are critical tools that enable breakthroughs in neuroscience research and the diagnosis/treatment of many neurological disorders like epilepsy and stroke. Another exciting technology that makes use of these devices is a brain–computer interface (BCI) or brain–machine interface (BMI). BCIs are restorative devices that aim to replace functionality an individual lost to neural injury or disease, and they demonstrate the variability and versatility of multichannel neural interfaces [1,2,3,4]. The methods of interfacing with the cerebral cortex and their corresponding electrodes can be mainly divided into four categories: external scalp recordings from electroencephalography (EEG), surface cortical recordings from electrocorticography (ECoG), surface cortical recordings from micro-electrocorticography (μECoG), and intracortical recordings from within the cortex and brain parenchyma using penetrating electrode arrays. Each type of neural interface methodology has its own advantages and disadvantages. EEG records neural signals through electrodes placed on the scalp. Due to its relative ease of use and non-invasive nature, EEG is a relatively well-known and commonly used method of acquiring neural signals. However, the information acquired from EEG is quite limited because the neural signal quality is diminished by the overlying tissues (i.e., scalp, soft tissues below the scalp, and bone) between the neuronal cells and the EEG electrodes. In contrast, ECoG electrodes are placed on the cerebral cortex, measuring local field potentials directly from the contact surface. This eliminates the attenuation/filtering of signals as they are transmitted through the skull and scalp, creating a more information-rich signal than EEG. However, conventional ECoG devices that are clinically available have an electrode site size of approximately 1 cm in diameter, which limits the spatial resolution of neural recording and stimulation [5]. 

Micro-ECoG electrode arrays utilize micro-scale electrodes with contact site diameters many orders of magnitude smaller than traditional clinical ECoG electrode sites and minimized inter-electrode spacing, allowing greater spatial resolution of the measured signals (Figure 1). Moreover, typical μECoG devices have ultrathin structure, thereby offering less invasive implantations [6]. Depending on the application, a μECoG device could have hundreds to thousands of electrode sites [7]. Lastly, intracortical electrode arrays can record individual action potentials from within the cortex or from deep brain regions. They give the most information-rich signal by recording individual action potentials in addition to intracortical local field potentials, but have the highest degree of invasiveness by penetrating into the tissue and eliciting an immune response to the foreign material. Among these device types, μECoG provides an appealing balance of information acquisition and spatial resolution with an acceptable degree of invasiveness (Figure 2). This article reviews the recent evolution of ECoG into μECoG, as well as the current direction these technologies are taking in the fields of engineering, neural interface research, and clinical medicine. Electrode array material choice is discussed, as is the role of ECoG and µECoG in the diagnosis and treatment of clinical disease pathologies, and current uses in BCI technologies, in addition to the host response to µECoG devices, in vivo imaging, and optical or electrical stimulation.

## 2. Evolution of ECoG into μECoG

Prior to the advent of µECoG, ECoG showed its advantages over EEG, providing greater temporal resolution than EEG, particularly regarding high gamma modulations (70–105 Hz) as the skull attenuates higher-frequency signals [5,8,9,10,11,12]. A gamma wave is a pattern of neural oscillation in humans or animals with a frequency between 25 and 100 Hz (and even above), although 40 Hz is typical. According to a popular theory, gamma oscillations are linked with high-level cognitive functions such as memory, attention, volitional movement, and conscious perception, which led to the theory that gamma activity plays a role in higher-level cortical processing [13,14]. Penfield was the first to describe the use of intraoperative ECoG to record localized abnormal neural activity in a seizure patient in 1947 [15]. Since this pioneering work, ECoG was used as a standard of care for clinical mapping of eloquent cortex prior to therapeutic resection of brain tissue. 

ECoG rests on the surface of the cortex and is, by nature, less invasive than traditional intracortical microelectrodes by eliciting less of an immune response, and demonstrates better signal longevity [16,17,18]. ECoG BCIs concentrate on both restoring communication [19] and the control of prosthetic limbs [20,21]. ECoG devices further demonstrated their clinical applicability for BCI control with chronic implantation in animal models [22,23] and acute implantation in humans [21,24]. Most notably, Shimoda et al. demonstrated the signal stability and longevity of ECoG for decoding of continuous three-dimensional (3D) hand trajectories in non-human primates over several months [25]. Equally importantly, Breshears et al. demonstrated that pediatric brain signals for hand movement could be easily and accurately decoded using ECoG (70–99% accuracy with ~9 min training) [26]. Chronic ECoG implantation in humans is currently being investigated as a treatment/warning system for epilepsy with limited success [27]. Chronic BCI testing using ECoGs to control devices is limited, as most subjects are epilepsy patients fitted short-term with ECoG for diagnostic reasons, which are then recruited into research projects [11,28,29,30]. The major caveat to this model is that, within these small experimental groups, there is huge variability in patient age and health status, as well as the site and number of electrodes implanted [31]. High-density ECoG was also used to decode motor imagery of sign language gestures as an alternative mode of communication [29]. 

Micro-ECoG is becoming increasingly popular for its ability to provide higher temporal and spatial resolution than typical ECoG [6,17,32,33], often comparable to intracortical microelectrodes [33,34]. A major advantage is the smaller size of the electrode site, which allows for more precise and accurate readings and less invasive implantation than its ECoG predecessor [17,33,34]. The reduction in invasiveness predominantly refers both to the reduced size of the craniotomy required, as well as the amount of bulk material that is implanted, regardless of its implantation site. Bundy et al. suggested that μECoG should be implanted subdurally to avoid reduction in signal amplitude in humans [35], but μECoG arrays can also be implanted atop the dura with only a slight loss in signal quality. Chronic epidural μECoG implants for BCI control were successfully demonstrated in non-human primates [36]. Micro-ECoG was also applied to read local field potentials from below the cortical surface. By applying a sparse linear regression algorithm to μECoG readings, Watanabe et al. demonstrated decoding of the hand trajectories in 3D space from depths of 0.2 to 3.2 mm, comparable to readings from more invasive microelectrode arrays [37]. Like ECoG, μECoG demonstrated applications for restoring communication and controlling prosthetic limbs. Kellis et al. demonstrated the effectiveness of μECoG to classify spoken words and distinguish between phonemes in humans [33,34]; however, these studies demonstrated limited success (<50% accuracy). The major caveat to decoding speech with μECoG relates to the spatiotemporal dynamics [30,38]. Speech involves a plethora of functional domains including motor, visual, auditory, and language domains in the high gamma range alone [30,36,38,39]. Brain activity becomes even more complex when the results are generated in real-life social settings, rather than the typical heavily controlled laboratory setting performing pre-determined tasks [28]. The continued research into decoding speech using both macro- and micro-ECoG is likely to be mutually beneficial. Alternatively, ECoG over the motor cortex is used for restoration of communication via BCI control of a computer cursor on a digital keyboard. Control of cursors on computers can be easily and rapidly learned with exceptional accuracy in both non-human primates [40] and humans [30,41]. Rouse et al. demonstrated that non-human primates could rapidly learn (days) to control velocity of a computer cursor with closed-loop recording of differential gamma-band amplitude (75–105 Hz) via μECoG [36]. 

## 3. Micro-ECoG: Electrodes and Substrates 

Each of the configurations mentioned above can utilize a wide variety of materials to obtain specific electrical recording types. These materials range from traditional biocompatible metals such as platinum, as well as new advances in the use of advanced two-dimensional materials such as graphene [42,43,44]. Not only do the materials themselves behave differently, but their properties can be further tunable via surface treatments or modifications. In this review, we categorize µECoG electrodes in terms of the electrode materials and review their usage. 

### 3.1. Platinum

Platinum is a common material used in various applications of neural stimulation and recording due to its ability to resist corrosion and its long history of biocompatibility in the brain. This allows for long-term reliability of electrodes to be used in chronic studies [42]. Also, platinum is common in general microfabrication due to the ease of its fabrication process, which makes it readily amenable to most electrode construction protocols [43]. 

Furthermore, platinum is able to inject current into the brain through reversible reactions limiting damage or harm to the cortex. This current injection is achieved through a combination of Faradaic and double-layer charging, with the Faradaic component being the driving force under most neural stimulation conditions [43]. This Faradaic component is primarily from a displacement current component of the injected current achieved when the electrode is behaving as a capacitor. 

These properties make platinum a viable material for use in many studies. One downside is that the materials are not transparent, which makes it impossible to do optical imaging of the cortex directly at the contact site [44,45]. Current uses for platinum electrodes include restoring or improving impairments in the visual, auditory, and somatosensory regions of the cortex through neural stimulations. With advances in technology throughout the field of neural engineering, improved platinum electrodes may show promise in prosthesis technology [42].

### 3.2. Sputtered Iridium Oxide

Iridium-oxide films are emerging as a technology in neural stimulation electrodes as a means to increase the electrode’s ability to inject charge. These electrodes are able to inject charge via reversible reduction and oxidation between Ir^3+^/Ir^4+^ valence states within the oxide film. By changing the thickness of the iridium-oxide layer, the electrical characteristics of the electrode can be tuned. This leads to a large variety of properties that can be obtained for the electrode [46]. 

One downside to iridium oxide is that it is more brittle compared to platinum, which prevents it from being used in flexible electrodes. This can prevent good contact with the cortical surface electrodes, as well as reduce the biocompatibility of the electrode due to the difference in the mechanical compliance of the electrode versus that of the brain tissue [43,45,47]. 

### 3.3. ITO

Indium tin oxide (ITO) is another potential candidate for transparent electrodes as it is used for commercial transparent electrodes in displays such as liquid crystal displays (LCDs) or active-matrix organic light-emitting diodes (AMOLEDs) [48,49]. Ledochowitsch et al. reported fabrication and characterization of a 49-channel ITO-based µECoG array [48]. Kwon et al. demonstrated an opto-µECoG array based on ITO epidural electrodes and integrated light-emitting diodes (LEDs) for optogenetics [49]. Due to the transparency of ITO (~80%), optical stimulation to brain tissue through the electrode was enabled. Kunori et al. demonstrated cortical electrical stimulation (CES) through ITO-based µECoG to investigate activation profiles of the cortex using a voltage-sensitive dye (Figure 3) [50]. CES is a technique that already reached clinical use in human patients through macro-ECoG devices. The implementation of CES through µECoG provides a useful tool in determining many of the effects that electrical stimulation has on the brain [51].

However, in vivo studies with ITO electrode arrays are limited to acute animal experiments. In fact, the brittleness, limited transparency near ultraviolet (UV) light, and process dependency of ITO appear to be the limitations in terms of chronic in vivo studies and the compatibility of specific neural imaging modalities [49,50,51]. 

### 3.4. Graphene

In recent years, optical imaging of the cortical areas of the brain while recording the electrical activity through surface electrodes became possible [44,52,53,54]. This is due to the availability of conductive, optically transparent materials, unlike conventional metal-based conductive materials. Graphene’s optically transparent nature and electrically conductive properties make it a good material for cortical electrode implementation. Graphene-based clear electrode arrays were used for a variety of optogenetic studies where light-evoked potentials could be measured on the same cortical areas that the light was administered [44]. Specifically, mouse species expressing light-sensitive proteins, either passed down genetically or through transfection, could undergo neuronal stimulation in the presence of certain wavelengths of light [55]. This makes clear µECoG appealing since it permits optical stimulation of the cortex directly below the recording site. This allows for more thorough probing of neural circuitry within the cortex, as well as other imaging modalities, simultaneously [44,54]. Generation of light-induced artefacts is one of the challenges in an integration of optical modalities with electrical recordings. However, this type of artefact could be minimized to enable cross-talk-free integration of two-photon microscopy, optogenetic stimulation, and cortical recordings in the same in vivo experiment [56].

In addition, graphene’s mechanical compliance may help improve the long-term biocompatibility of the electrode. It is reported that graphene electrodes remain viable for chronic recording for extended time periods (70 days) [44].

In most cases, µECoG electrode electrical properties can be modeled by a constant phase element (Z_CPE_), Warburg impedance (Z_W_), charge transfer resistance (R_CT_), and a solution resistance (R_S_), as presented in Figure 4 [57]. Graphene’s high transmittance and low electrical impedance make it a prime candidate for optically clear electrodes. According to Li et al. (2009), improved graphene development processes can make graphene sheets with low resistances. Similarly, graphene is able to achieve transmittances above 96% for single-layered graphene between the wavelengths of 400 nm and 1000 nm [58]. Park et al. characterized optical transparency of a four-layer graphene electrode at ~90% transmission over the ultraviolet-to-infrared spectrum, and demonstrated its utility for use in in vivo imaging and optogenetics (Figure 5) [44].

Graphene electrodes, like most electrodes, are electrically characterized with a resistive–capacitive model [54,59]. Therefore, as in all biological/electrical interfaces, resistance of the electrode changes with frequency. Typically, electrodes are characterized by this frequency response. Neural electrodes are also commonly characterized by their resistance at 1 kHz. [59] This is a common benchmark for neural electrodes due to the fact that the fundamental frequency of the neural action potential is at this frequency. While electrodes are typically characterized by their impedance at 1 kHz, this impedance can be quite variable, and ranges in vivo from approximately 50 kΩ to 1 MΩ [44,54,60], depending on the site size and material. 

Previously reported µECoG devices are summarized in Table 1. For reference, penetrating electrode works are also summarized in Table 2. 

### 3.5. Bioresorbable Silicon

In clinical neurological monitoring involving µECoG with the abovementioned materials, a second surgical procedure for removal of the device is typically performed after the recording is over. Whether the implant is extensive or not, such a second procedure often adds cost and risk. In most cases, one to three weeks of recording is required. Ideally, a temporary monitoring system that can dissolve or disappear after the suggested period of implant time would eliminate such a second surgical procedure. Recent advances in silicon devices demonstrated bioresorbable forms of silicon sensors and electronics, where ultrathin silicon nanomembranes disappear after a certain period of time in fluids. For instance, a hydrolysis demonstration of block silicon nanomembrane (initial dimensions: 3 mm × 3 mm × 70 nm) in phosphate-buffered saline (PBS) at 37 °C suggests that a complete dissolution occurs after 12 days. It was also demonstrated that the constituent materials comprising such bioresorbable sensors and electronics are biocompatible, which is suitable for biomedical applications [70].

Precise recordings of brain signals from the cerebral cortex were achieved utilizing bioresorbable silicon electronics [69]. With an array of electrodes made of silicon nanomembranes mounted on bioresorbable poly(lactic-*co*-glycolic acid) (PLGA) substrate, the flexible µECoG device could achieve conformal contact with the cortex, owing to the ultrathin structure of the device. This technique of utilizing bioresorbable substrate was also demonstrated with traditional metal electrodes where the conformability of the electrodes was improved by eliminating the normal substrate, such as polyimide, and replacing it with bioresorbable silk [68]. Furthermore, sophisticated bioresorbable silicon µECoG arrays with actively multiplexed electronics involving silicon transistors were demonstrated for large array-based spatial mapping of cortical activity. The multiplexed electrode array using flexible silicon electronics was proven to achieve extremely high density (up to 25,600 channels) for precise mapping of the brain activity. Such a concept provides a robust foundation for bioresorbable implantable electrode technology, especially as the use of silicon aligns well with mature semiconductor manufacturing infrastructure [7].

Drawing from the Table 1 and Table 2, a multitude of different studies can be formulated. Overall, the use of different materials within the microarrays is still up for debate, and wide varieties are still in testing. Additional materials such as graphene and poly(ethylenedioxythiphene) (PEDOT) were added to the traditional materials. These vary greatly from the traditional metallic electrodes in composition, but strive to imitate the electrical characteristics that are desirable [60]. In all cases, the general characteristics are known, but with each material having its own specific drawbacks. Overall, neuro-recording and stimulation are emerging fields, as a greater understanding of brain processes is required. Given this push, along with precise manufacturing techniques, the variety of implementation will go up. However, until long-term studies can be completed, the use of the original metallic electrode microarrays (Pt, Ir, and Tn) will remain the clinical standard. 

## 4. Host Response to µECoG Devices

The brain has a unique and complex response to trauma that is heavily mediated by neurogenic inflammation. The complex inflammatory response to brain injury following trauma can be neuroprotective, but can also result in secondary injury, driving chronic neural injury. Neurogenic inflammation in response to trauma is beyond the scope of this review, and was best described elsewhere [47,82,83,84]. Of particular interest to this review is the chronic foreign body response (FBR), as implanted electrodes often incite an FBR, which can both affect the performance of implanted electrodes and the surrounding brain tissue itself [47,85,86]. 

Whilst host cells immediately respond to the surgical injury itself, the foreign body (electrode) induces chronic inflammation at the biotic–abiotic interface [47,87]. At the biotic–abiotic interface, microglia (resident immune cells of the central nervous system (CNS), analogous to macrophages in the rest of the body) become activated, undergo gliosis, and eventually encapsulate the implanted device [47]. The primary cause of this reaction is yet to be elucidated; however, the strongest evidence indicates that a mismatch between implanted materials and tissue compliance heavily mediates the activation of microglia, as demonstrated by several eloquent in vitro studies [88,89]. Increasing evidence demonstrates the importance of material properties on cell fate, including neural stem and progenitor cells, which holds implications for neural regeneration around the electrode site [90]. 

The most invasive electrodes, such as penetrating electrode arrays, cause the most trauma at the time of implant, and also elicit the greatest FBR response as a result of increased surface area between the implanted foreign body and the native tissue [85,86]. In contrast, less invasive devices, such as µECoG, are thought to generally elicit less of a response, demonstrated by greater electrode longevity [66,67,91]. 

Most commonly, implanted devices (particularly penetrating devices) become encapsulated in a glial scar similar to macrophage-induced fibrosis in other organs [92]. The foreign body response is dynamic, and considered an evolutionary survival mechanism to either remove or compartmentalize foreign objects (not self), preventing their interaction with surrounding tissues (self) as a means of self-preservation. The glial scar, astrocytes and microglia responding to a foreign body, can isolate the electrode from the desired neurons and insulate it from the rest of the cortex. This can lead to an increase in impedance, and make it harder for the electrode to record the electrical activity of the underlying tissue [47,59,85,86]. Astrocytes can be identified by increased expression of glial fibrillary acidic protein (GFAP) and vimentin [93,94]. Microglia are often identified by immunostaining for ionized calcium-binding adaptor molecule 1 (Iba1). Glial scars consist of an excess of extracellular matrix, including collagen IV and chondroitin sulfate proteoglycans [95]. The increase in inflammatory cell density and extracellular matrix deposition both lead to increased impedance and decreased recording capability [59].

Aside from the cellular elements of scarring, molecular elements such as proteins are known to adhere to the surface of recording sites (biofouling). These protein layers typically have no reactive impedance on signals below 5 MHz. Therefore, the buildup of protein can be modeled as an increase in series electrolytic resistance in the equivalent circuit. The electrode–electrolyte interface impedance is comparable to that of a high-pass filter, with larger impedances for low-frequency signals. This increase in electrolytic resistance increases the impedance for signals of all frequencies. This causes an upward shift in the virtual cutoff frequency, making the device more susceptible to noise at lower frequencies, and decreases the amplitude seen by the amplifier circuit, lowering the signal-to-noise ratio (SNR). Electrode design factors such as geometry, materials, and level of invasiveness all play important roles in the longevity of electrodes by reducing glial scar formation and biofouling. Providing open space as opposed to solid electrodes was shown to reduce scar formation [77,96]. Reducing invasiveness (µECoG vs. penetrating arrays) may also reduce scaring through reducing trauma, both to the parenchyma and the blood–brain barrier [67,87].

The host response can significantly affect the performance of the electrode. Typically, the implanted neural electrodes show a large increase in the impedance of the electrode after implantation for the first 7–10 days [44,59,65]. This is speculated to primarily be due to the host response to the implantation surgery rather than electrode degradation. 

## 5. Role of ECoG and µECoG in Human Disease and BCI

The role of macro- and micro-ECoG for the clinical treatment of human patients is expanding. Seizure focus localization is the major traditional role for ECoG clinically [97,98]. Intraoperative ECoG can be used to identify abnormal interictal discharges as a proxy for the epileptic focus, but numerous constraints, especially limited time, make identification of a seizure focus in the operating room unreliable. Instead, temporarily implanted subdural ECoG arrays, often in conjunction with depth electrodes, provide longer-term monitoring, during which withdrawal of antiepileptic drugs and recording of multiple seizures can help localize the region of seizure onset [11,18,97].

In addition to localizing the source of seizures, ECoG can also be used to localize the eloquent cortex that must be spared during surgical resection. Traditionally, this is achieved with intraoperative mapping via bipolar cortical stimulation and identification of corresponding motor/sensory response or speech arrest, with ECoG arrays utilized to monitor for stimulation-induced after-discharges, which raise concern for stimulation-induced seizures. Performing eloquent cortex mapping with stimulation via an implanted ECoG array outside the operating room removes time constraints and results in a more detailed functional map. Cortical mapping using implanted ECoG arrays outside the operating room also negates the need for awake surgery, a key concern to maximize patient comfort especially for those patients unable to tolerate awake surgery [8,9,11,18,30,99,100].

The unpredictability of seizures is one of the sources of morbidity in epilepsy. If a patient has some warning of an impending seizure, they may be able to prepare for the event by making modifications to their physical environment or medication dosing. An implanted subdural ECoG array (NeuroVista, Seattle, WA, USA) linked to a subcutaneously implanted battery and telemetry unit that communicates with a patient advisory device was used to provide patients with an early warning of a possible impending seizure, with promising results reported in 2013 in an early feasibility human trial involving 15 patients [101].

The ability to detect impending seizure activity also opens the possibility of potentially interrupting that activity with direct responsive neurostimulation (RNS). In patients with seizure foci that are not amenable to surgical resection (e.g., foci involving the eloquent cortex or bilateral hippocampi), responsive neurostimulation (NeuroPace, Inc., Mountain View, CA, USA.) was shown in a randomized multicenter double-blinded controlled trial involving 191 patients to significantly decrease the frequency of partial-onset seizures, with a median reduction of 53% at two years. This system utilized either ECoG strip electrodes (1 × 4) or depth electrodes to provide continuous monitoring of electrical activity with subsequent stimulation based on specific abnormalities associated with seizure onset [102,103].

In an investigative fashion, subdural ECoG was used in humans to evaluate cortical activity surrounding areas of brain injury in patients with ischemic stroke, traumatic brain injury, and aneurysmal subarachnoid hemorrhage. These studies demonstrated frequent episodes of cortical spreading depolarization and depression around the area of injury and the resultant increased metabolic demand was associated with neurological worsening. It is uncertain at this time whether interventions based on detecting these episodes of cortical spreading depolarization or preventing them can be used to improve clinical outcomes [104,105,106,107].

Cortical stimulation via ECoG, coupled with rehabilitation therapy, was also postulated to aid functional recovery after stroke. Despite promising animal studies [57,108,109,110,111,112,113] and early human trials [114,115,116,117], a large multicenter randomized controlled human trial using a fully implanted epidural ECoG array and battery (Northstar Neuroscience, Inc, Seattle, WA, USA) to deliver continuous stimulation over an area of chronic infarct, combined with intensive therapy, failed to demonstrate clinically significant benefit [118,119].

The application of BCI for the control of prosthetic limbs exploded in the last decade, predominantly encouraged by the Defense Advanced Research Projects Agency’s (DARPA) Revolutionizing Prosthetics Program [120]. Several groups demonstrated various applications for μECoG in the decoding of upper limb movements for control of prosthetics by humans, including virtual hand opening and closing [32], finger movements [6], and wrist movements [121]. Leuthardt et al. demonstrated that μECoG can be used to identify and separate motor movements in the wrist from <5 mm of motor cortex in humans [121], whilst Wang et al. showed that μECoG can be used by a patient with tetraplegia to control a cursor on a computer in both two and three dimensions [122]. Micro-ECoG is yet to be tested for the range of applications of its macro predecessor, such as for controlling the latest multifaceted, modular upper prosthetic limbs [123]. 

## 6. Discussion and Future Direction

The development of multichannel neural interfaces, including µECoG, allowed for great advances in understanding the link between neural activity and body function, as well as exploring the cause of neurological disorders such as epilepsy. Furthermore, these technologies enable the development of neuroprosthetic devices and therapies that hold tremendous potential to restore an individual’s motor and sensory function that was lost to disease or traumatic injury. Due to its balance between invasiveness, spatial resolution, and biocompatibility, µECoG is a technology that is ideally placed to provide stable, reliable neural interfaces for years to come in both the research and clinical domains. 

The use of µECoG in basic science and pre-clinical research gained significant momentum over the past decade especially for work exploring brain–computer interfaces and examining the viability of cortex following neural injury. However, µEcoG is still not the preferred method for recording cortical neural activity in the majority of neuroscience research, as it struggles to isolate the spiking activity of individual neurons, especially those from deeper cortical layers. The relationship between spiking activity in deeper cortical layers and the signal recorded by µECoG is an active area of research for many in vivo biophysical and computational modeling studies. We expect that results of these experiments could result in new techniques for localizing and predicting individual sources of neural activity leading to the greater usage of µECoG technology. 

Despite the potential of µECoG alone, we see its greatest potential when used in concert with optical stimulation/imaging techniques to dissect the function of neural circuits. As we discussed above, the development of optically clear µECoG electrodes enables the simultaneous recording of cortical potentials and neural stimulation via optogenetic techniques (see Figure 5). More exciting is the combination of optically clear µECoG electrodes, advanced optical imaging modalities (i.e., multiphoton imaging and light sheet microscopy), and animal models with genetically encoded sensors offering the opportunity to interrogate structures located farther from the cortical surface. These techniques offer the ability to explore the relationship between electrophysiology, cellular metabolism, and vascular dynamics, which will be necessary to understand the etiology of many neural diseases like epilepsy.

Although μECoG shows great promise for clinical application, it has yet to reach widespread utilization in the diagnosis and treatment of human disease. The underwhelming use of μECoG in clinical settings stems from two factors. Firstly, there is currently no Food and Drug Administration (FDA)-approved device/indication for μECoG. While challenging, gaining the approval of regulators seems a matter of time given the similarity of μECoG with its technological cousin, ECoG. We expect that this obstacle will be overcome in the near future. Secondly, there is currently no pressing clinical need for μECoG, as current-generation ECoG technology satisfies today’s clinical usage. For example, the use of ECoG in epilepsy patients drove much of what is currently known about the functional organization of the human cortex. We expect the use of µECoG to further push the boundaries established by conventional ECoG due to its ability to measure more detailed electrophysiological data, and the less invasive nature of µECoG has the potential expand the patient population appropriate for implanted devices. We believe that the application of μECoG for BCI will soon replace macro-ECoG as the new standard, due to its higher spatio-temporal resolution and reduced manufacturing limitations.

Aside from replacing current-generation ECoG with µECoG, new clinical indications requiring µECoG are on the horizon. Implanted devices, such as deep brain stimulation for movement disorders and responsive neurostimulation for the treatment of epilepsy, moved out of labs and are now standard-of-care treatment for thousands of patients. Micro-ECoG will most certainly be used to add closed-loop stimulation capabilities to the future generation of neuromodulation devices. Here, μECoG could provide a richer stream of electrophysiological information that will fine-tune decisions regarding when and where to initiate therapeutic stimulation. Furthermore, μECoG could most certainly be used to monitor neural signals in the first-generation clinical neuroprosthetic devices due to its relatively high spatial resolution and biocompatibility. However, the clinical viability of neuroprosthetic devices hinges on improvements in the wireless transmission of data and power, which will allow for a fully implantable form factor. We see a potentially bright future for μECoG technologies, one where many patients will see benefits from future generations of implanted neuromodulation and neuroprosthetic devices. The use and utility of µECoG is clearly ascending, as its core and supporting technologies are being refined and new applications are being imagined. Next-generation technologies could be catalyzed by the development of μECoG devices that are fully untethered from the external world and include all of the necessary electronics (i.e., data acquisition, power transmission, and communication) directly on the device [124,125,126,127]. Such developments will enable integrated neuroprosthetic and neuromodulation systems that will have the ability to function for the lifetime of the patient. 

## Figures and Tables

**Figure 1 micromachines-10-00062-f001:**
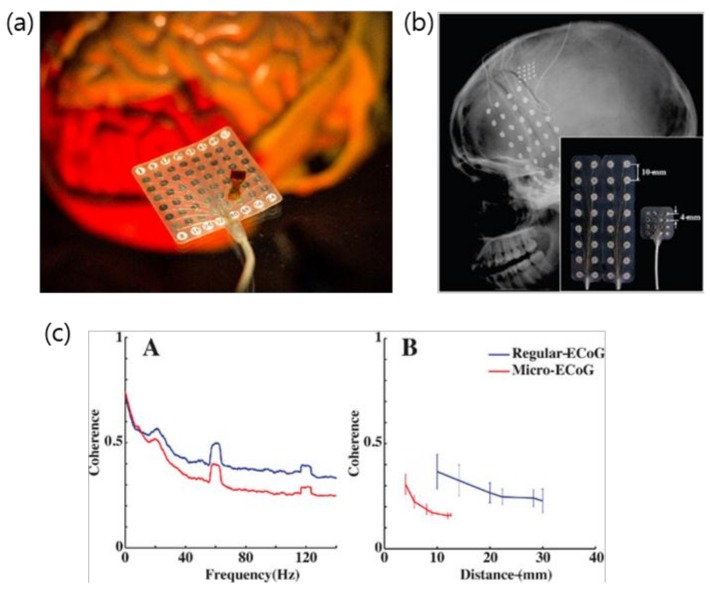
(**a**) Picture of a clinical electrocorticography (ECoG) grid underneath a micro-ECoG (μECoG) array. Side-by-side comparison of the regular macro-ECoG and μECoG arrays showing difference in electrode spacing. (**b**) X-ray image showing the implanted ECoG and μECoG electrode. (**c**) Coherence analysis to characterize independent neural signals recorded from both macro-ECoG and μECoG. This suggests μECoG offers higher spatial resolution for neural signal recording. (**a**) Photo was taken at Neural Interfaces Research (NITRO) lab at University of Wisconsin (UW) Madison; (**b**,**c**) reprinted with permission from Reference [6].

**Figure 2 micromachines-10-00062-f002:**
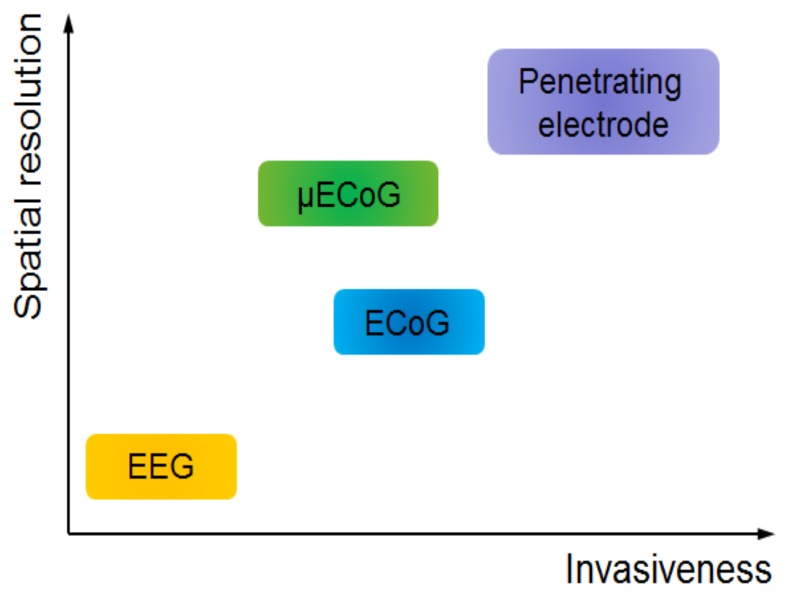
Spatial resolution versus invasiveness for various types of neural electrodes. Micro-ECoG has a balanced spatial resolution and invasiveness.

**Figure 3 micromachines-10-00062-f003:**
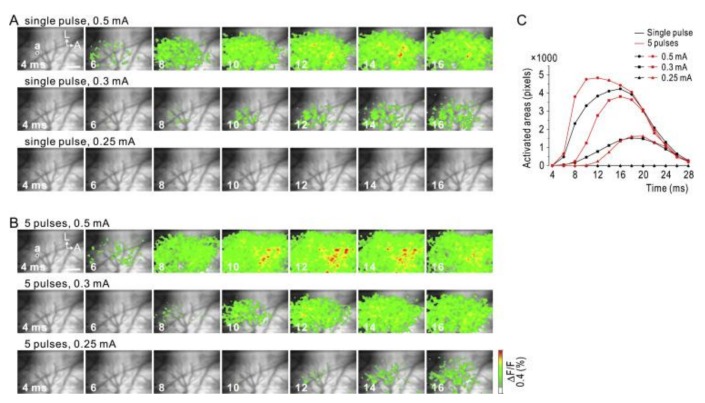
Anodic stimulation via indium tin oxide micro-ECoG. Neural activity captured via fluorescent voltage sensitive dye. (**A**) The white circle (a) indicates a clear electrode used for stimulation. Activation profiles captured after delivering single pulses of current intensity of 0.5, 0.3, and 0.25 mA. (**B**) Duplicate of experiment in (**A**) with a pulse train of five pulses at 500 Hz. (**C**) Comparison of spatial activation spreading due to different stimulation settings. The spatial extent of activity was evaluated by the number of pixels above threshold. A, anterior; L, lateral. Scale bar, 1.0 mm. Reprinted with permission from Reference [50].

**Figure 4 micromachines-10-00062-f004:**
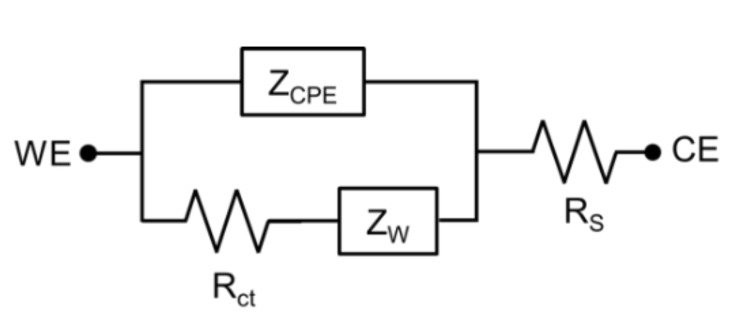
The representative equivalent model of a µECoG electrode. WE, working electrode; CE, counter electrode; Z_CPE_, constant phase element; Z_W_, Warburg impedance; R_CT_, charge transfer resistance; R_S_, solution resistance.

**Figure 5 micromachines-10-00062-f005:**
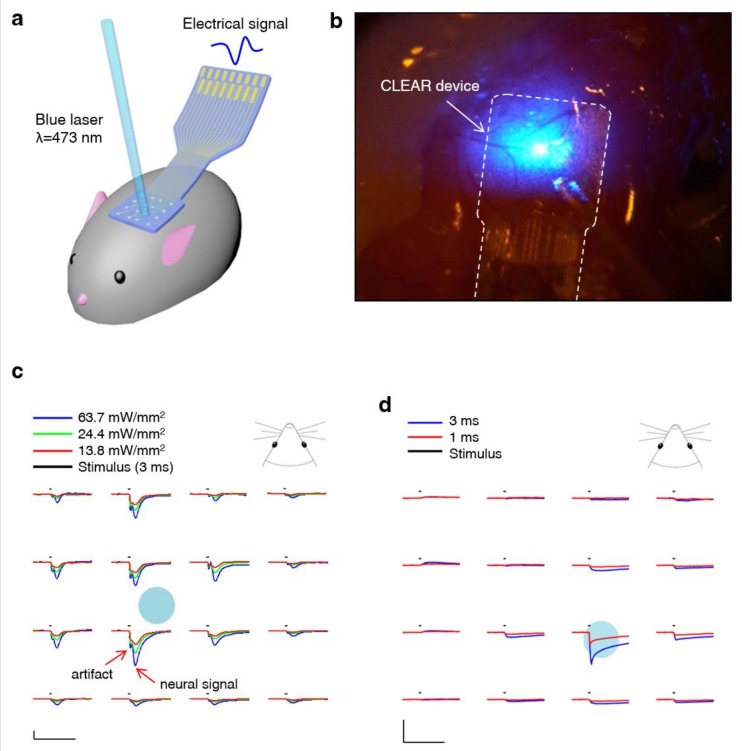
(**a**) Illustration depicting experimental ensemble combining optical stimulation with µECoG in a mouse model. (**b**) Optical illumination and stimulation spatially control over the mouse brain and µECoG via an optical fiber. (**c**) Spatial mapping of local field potentials obtained from a graphene µECoG throughout an optically evoked potential on the cortex of a channel rhodopsin positive mouse; *x*-scale bars represent 50 ms, *y*-scale bars represent 100 μV. (**d**) Post-mortem control depicting photo-electric artefact generated during blue-light optical stimulation; *x*-scale bar, 50 ms; *y*-scale bars, 100 μV. Reprinted with permission from Reference [44].

**Table 1 micromachines-10-00062-t001:** Comparison of different electrocorticography (ECoG) and micro-ECoG (µECoG) electrodes with regards to various parameters.

Layout	Substrate Materials	Recording Site Materials	Size/Impedance	Notes	Reference (Year)
2D planar array	Polyimide	Pt	1 mm^2^ 1.5–5 kΩ	255 channels LFP and ECoG recording awake monkey for 4 months	[22] (2009)
2D planar array	Parylene C	Au-PEDOT:PSS	10 × 10 µm^2^ 0.2 MΩ	LFP and ECoG recording in freely moving rat and humans 256 channels	[61] (2015)
2D planar array	Parylene C	Graphene	Diameter: 150–200 µm, 100–600 kΩ	Transparency evoked potential by light (Optogenetics)(lifetime >70 days)	[44] (2014)
	Parylene C	Pt	Diameter: 150–200 µm 50–300 kΩ	(lifetime >70 days)	
2D planar array	Silicone rubber	Pt	-	SEP recording (µECoG) and stimulation	[62] (2011)
2D planar array	Parylene C	Sputtered indium tin oxide (ITO)	49-channel (Pitch of 800 μm ) 16-channel (Pitch of 200 μm)	Design, fabrication, and characterization	[48] (2011)
2D planar array	Parylene C	Sputtered indium tin oxide (ITO)	Diameter: 200 µm 100–200 kΩ	Optogenetics with integrated LEDs	[49] (2013)
2D planar array	Polyimide	Au-PEDOT	100 µm × 100 µm ~2.1 kΩ	recording from rat somatosensory cortex in vivo	[63] (2015)
2D planar array	Parylene C	PEDOT:PSS	10 × 10 µm 210–50 kΩ	Spike recording from surface (NeuroGrid),256 channel	[61] (2015)
2D planar array	Polyimide	Pt	300 × 300 µm^2^ ~20 kΩ	Multiplexing with integrated transistors Electrographic seizures	[7] (2011)
2D planar array in a chamber system	Polyimide	Au	Diameter: 200 µm 24–45 kΩ	124-channel µECoG and 32-channel microdrive, Multi-unit, LFP, µECoG comparison	[64] (2015)
2D planar array, perforated	Parylene C	Pt	Diameter: 200 µm	16 channel, optimizing vascular imaging.	[65] (2013)
2D planar array	Polyimide	Pt and Au	Diameter: 300 µm 5–10 kΩ	32-channel µECoG	[66] (2011)
2D planar array	Parylene C	Pt	Diameter: 200 µm<1000 kΩ	16 channel µECoG arrays, varying array footprint.	[67] (2014)
2D planar array	Silk	Au	30 electrodes	Mesh structure for conformal contact	[68] (2010)
2D planar array	Polyimide	Pt	360 channels each electrode 300 um × 300 um	Multiplexed using Si transistors	[7] (2011)
2D planar array	PLGA	Si	256 channels overall 3 cm × 3.5 cm	Bioresorbable	[69,70] (2016/2012)

2D, two-dimensional; Pt, platinum; Au, gold; Si, silicon; LED, light-emitting diode; LFP, local field potential; PLGA, poly(lactic-*co*-glycolic acid); PEDOT, poly(3,4-ethylenedioxythiophene); PSS, poly(styrenesulfonate); SEP, somatosensory evoked potential.

**Table 2 micromachines-10-00062-t002:** Comparison of different penetrating electrodes with regards to various parameters.

Electrode Type	Layout	Substrate Materials	Recording Site Materials	Size/Impedance	Notes	Reference (Year)
Micro wire	3D array	N/A	Stainless	50 µm ×50 µm 64 channels	Primary auditory cortex (rat, ECoG recording)	[71] (2006)
	3D array	N/A	Stainless Or Tungsten	50 µm × 50 µm Teflon coated	Single cortical neurons (monkey)	[72] (2003)
	3D array	N/A	Tungsten	35 µm^2^	Cerebral cortex (rat)	[73] (1999)
Michigan	Assembled 3D array	Si	Ir	100 µm^2^, 2 MΩ	LFP	[74] (2000)
Michigan	Assembled 3D array	15 µm thickness of Si	Ir	177 µm^2^, 0.72 MΩ 312 µm^2^, 1.65 MΩ	Cerebral cortex (rat) Chronic recording (127 days)	[75] (2004)
Michigan	2D array	Si	PEDOT & Au	Gold, 9.1 MΩ PEDOT, 0.37 MΩ	Single unit implanted in layer V (rat)	[76] (2011)
Michigan	2D array	Si	PEDOT	-	PEDOT VS Carbon A new set of materials to make fundamental Chronic single unit spikes in cortex	[77] (2012)
Utah	10 × 10 3D array	Doped Si	Ti/Pt (50/240 nm)	Width 80 µm, length 1500 µm	Insulated with polyimide	[78] (1992)
Utah	10 × 10 3D array	Doped Si	Pt/Ir	100–300 kΩ	Tip exposed (500 µm) Cat auditory & visual cortex	[79] (1999)
Utah	10 × 10 3D array	Doped Si	Pt/Ir	1600 µm^2^ 100–750 kΩ	Tip exposed (40 µm) Primary motor cortex (M1, monkey)	[80] (2005)
Utah	10 × 103D array	Doped Si	Pt	125 kΩ 2 mC·cm^−2^	Cortical stimulation/recording (>90 days in vitro)	[43] (2010)
		Doped Si	Sputtered iridium oxide film (SIROF)	6 kΩ 0.3 mC·cm^−2^	Cortical stimulation/recording (>90 days in vitro)	
Utah	Unrestricted freedom in the 2D probe	300 µm thickness of Si	Ti/Au/Pt (30/200/100 nm)	1–2 MΩ	72 channels Recording LFP in layers 1, 2, and 3 for 15 days	[81] (2009)

3D, three-dimensional; Ir, iridium; Ti, titanium.

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
