# Peer review of "Progress in the Field of Micro-Electrocorticography"

_micromachines, 2019, doi:10.3390/mi10010062_

Round 1

Reviewer 1 Report

This is an interesting review regarding progress in the field of micro-electrocorticography. There are however some issues that need to be addressed by the authors:

- A description of the methodology of this review is mandatory.

- The information provided in the "aknowledgement" section should be transferred to the "declaration of interest" section.

- All abbreviations should be explained at their first mention and then used accordingly throughout the text (as well as in abstract).

- Parentheses are missing from "Figure 1".

- A brief explanation of what "gamma modulations" is would be helpful for the non-expert reader.

-Lines 45-46: replace "nervous system" with "cerebral cortex"

- Lines 60, 142, 154, 190, 194, 256, 269, 337, 363, 370,  : references should be cited.

- Figure 3: "a" is not defined in the legend.

- Tables 1, 2: The abbreviations should be explained.

- The lines 387-388 should be transferred to the discussion section.

- Line 391: replace "organismal behavior" with "body function"

- Line 392: replace "for many pathologies" with "of neurological disorders such as epilepsy"

- Line 396: add the word "potentially" at the beginning

- Line 404: replace "will" with "could"

- Line 415: remove "and Alzheimer’s Disease"

- Lines 431, 434, 442: replace "will" with "could"

-Line 438: add "potentially" before "bright"

Author Response

Please see the attached response letter.

Reviewer 2 Report

uECoG is a promising neurotechnology for both pre-clinical and clinical applications. The authors provide an excellent overview on this technology, by discussing the comparison with other modalities, a history of this technology, different types of uECoG electrodes, foreign body responses, and clinical applications. This review article will help a wide range of scientists to contribute to this important niche area.

By addressing the following issues, the manuscript will be improved further.

1.       In the Abstract, it is slightly redundant to abbreviate several technical terms multiple times. For example, brain computer interface (BCI) (lines 19 and 31) and electrocorticography (ECoG) (lines 17 and 27).

2.       The reference format is odd. For example, in line 45, reference number was started from 4, not 1.  In lines 332-3, it looks a typo. In lines 372-373, it is wrongly formatted. In line 381, a space is not required after [117]. Please correct all of them.

3.       In Section 3.4., please mention about Thunemann and his colleagues (Nat Comm 2018) who did an excellent job to minimize optical artifact. Particularly because uECoG measures field potentials alone, optical artifact introduces a serious issue with the combination with optogenetics.

4.       In section 4, there seems to be a confusion about “activated glia.” In particular, in lines 308-9, the authors mention that GFAP is a marker of activated glial cells. But GFAP is a marker of astrocyte although astrocyte also contribute to foreign body responses and neuroinflammation. Microglia is often identified by immunostaining for Iba1. Please do more literature review on this and clarify this section.
